# Response of Diverse Peanut Cultivars to Calcium Fertilization under Clay Soil Conditions

**Mohamed Hamza [1], Mohamed Abbas [2],\*, Shimaa Sharaf [3] and Fatma H. Galal [1]**

[1] Department of Biology, College of Science, Jouf University, Sakakah 72388, Saudi Arabia; mhabdelhameed@ju.edu.sa (M.H.); fhgalal@ju.edu.sa (F.H.G.)

[2] Department of Natural Resources, Faculty of African Postgraduate Studies, Cairo University, Cairo 12613, Egypt

[3] Landscaping Unit, Supreme Council of Antiquities, Ministry of Tourism and Antiquities, Cairo 11534, Egypt; shaimaafarouk1985@gmail.com

\* Correspondence: msaelsarawy@cu.edu.eg

**Abstract:** The absence of calcium decreases the production of peanuts compared to any other element. To investigate the influence of calcium (Ca) applications on the production and seed quality of four diverse peanut cultivars from different regions—USA (NC-7), Egypt (Giza-6) and Nigeria (Samnut-23 and Samnut-24)—under clay soil conditions, two experiments were carried out at the Agricultural Experimental and Research Station, Cairo University, Egypt, during the two seasons of 2016 and 2017. The experimental design was a randomized complete block in a spilt-plot arrangement with three replications. The main plots were allocated to four peanut cultivars (Giza-6, Samnut-23, Samnut-24 and NC-7), and the sub-plots were devoted to calcium applications (soil application in the form of calcium sulfate dihydrate, foliar application in the form of calcium oxide and the control treatment of distilled water). Results indicated that all four peanut cultivars responded differently to the application of calcium fertilizers. The calcium application significantly enhanced peanut growth, yield components, biological, pod, seed, oil yields, seed oil, free fatty acids and seed calcium percentages in different cultivars. Soil calcium application significantly improved peanut production compared to foliar calcium application. NC-7 cultivar treated with the soil Ca application resulted in the maximum values of biological yield (92.9-ton ha$^{-1}$), pod yield (6.8-ton ha$^{-1}$), seed yield (4.4-ton ha$^{-1}$), oil yield (2247.0 kg ha$^{-1}$), pod index (203.2 g) and seed index (84.1 g). The interaction between the NC-7 cultivar and soil calcium applications is recommended to attain the best combination, leading to the highest yield and seed quality of peanuts.

**Keywords:** groundnut; growth; pod yield; oil yield; fatty acid; Ca; quality

## 1. Introduction

Peanut or groundnut is known as a "King of oilseed" [1]. It is one of the most important cash legume crops cultivated mainly for its edible seeds [2]. It is the thirteen most important food crop and the third major oilseed crop in the world [3]. Peanuts are a self-pollinated herbaceous plant belonging to the family *Fabaceae*; only *Arachis hypogaea* L. is domesticated and commonly cultivated [4]. Peanuts are a perfect source of vitamins, minerals, fats, protein and carbohydrates [3].

Calcium (Ca) is an essential nutrient found in soil that plays an important function in many biochemical processes [5]. Ca plays a significant role in the cell wall, membrane stabilization, cell expansion and division, cation and anion balance, osmoregulation and modification of certain enzymes [6]. Because calcium is a structural component of the cell wall, a regulator of cell homeostasis, an enzyme activator and participates in ion absorption, it plays a significant role in plant growth and yield [7]. In its ionic form, calcium acts as a structural component in plant cells as well as a key signaling molecule involved in multiple signal-transduction pathways [8,9]. Also, as a second messenger, calcium ties

environmental and developmental signals to the physiological response of plants [6]. Ca is mainly stored in the intercellular spaces, cell walls, vacuoles and organelles such as the mitochondria and chloroplasts of plant cells [10,11]. To maintain a good output in peanut yield, calcium supplementation should be used. [12]. Calcium fertilizers can significantly increase the rate of germination, growth, yield and quality traits of groundnut [13–15]. However, Kadirimangalam et al. [5] indicated that the addition of Ca before the pod development in peanut is essential to have enough calcium available to the pods, which could help in yield enhancement. Ca is an important factor in the peanut seed quality, especially in soils suffering from calcium deficiency [16]. Calcium is especially important in the production of peanuts and is required from the time when pegs begin to appear until the pods are mature [17].

Gypsum ($CaSO_4$) and calcium oxide ($CaO$) forms are available for soil and foliar applications to supplement calcium. Gypsum is one of the most cost-effective forms of calcium fertilizer for groundnuts, and it is essential for pod formation [18]. Several studies have demonstrated the significance of various calcium types on peanut development and productivity. Gypsum is the primary fertilizer used in peanuts, and the addition of gypsum significantly improves groundnut production [19,20]. However, the application of foliar calcium can provide an excellent opportunity to enhance seed production and oil quality [21]. Most of the pod calcium needs are taken directly from the soil, known as the fruiting area, through the shell of pods, rather than via the roots, and then down through the peg [22]. In the leaves and stems, higher Ca levels may influence the physiological and biochemical processes of groundnut, hence, indirectly influencing calcium absorption from the soil [23]. The foliar calcium application restored temperature-dependent photosynthesis feedback inhibition due to improved growth/sink demand [24]. Ca foliar application is a strategy that farmers are increasingly employing to augment fertilizer in the soil [25]. Calcium treatment improves development, leaf nutrient, productivity and quality attributes of groundnut [26]. Also, spraying Ca increased leaf development and dry matter accumulation in peanut roots, stems and leaves [24]. It is often assumed that foliar calcium fertilizer treatment has a minimal effect on enhancing peanut production [15]. Likewise, foliar application is not considered an efficient strategy to prevent Ca deficiency in peanuts when compared to the soil application of calcium fertilizer [15,27].

Calcium deficiency decreases peanut production and quality traits more than any other nutrient. Due to a shortage of calcium in the pegging zone, the pegs formed very few pods. Kamara et al. [28] indicated that the calcium application produced the highest filled pods. Calcium shortage results in a rising number of empty pods, poor quality seeds, black shaded seeds plumule and a high prevalence of pod rot, resulting in a severe loss of production and a lower grade [16,29,30]. Groundnut pops (full size pods with no seeds inside) or pods with only one seed, darkened plumules in the seed, poor germination and pod rot are considered big problems facing peanut producers and exporters, and a severe lack of calcium in the podding zone is the main cause of pops. In the present investigation, the role of Ca applications in peanuts will be discussed under clay soil conditions. There is little information available regarding the performance of Giza-6, Samnut-23, Samnut-24 and NC-7 cultivars grown in clay soil with calcium deficit. Thus, the objective of current study was to investigate the effect of soil and foliar applications of calcium on the productivity and seed quality of four diverse peanut cultivars from different regions—USA (NC-7), Egypt (Giza-6) and Nigeria (Samnut-23 and Samnut-24)—under clay soil conditions. The obtained results can provide useful information to identify the best calcium fertilizer application based on the production of peanut cultivars under clay soil conditions.

## 2. Materials and Methods

### 2.1. Experimental Site

The experiments were carried out at the Agricultural Experimental and Research Station, Faculty of Agriculture, Cairo University, Giza, Egypt (30°02′ N and 31°13′ E, altitude of 30 m above sea level) during the two successive seasons of 2016 and 2017.

Monthly mean temperature, monthly relative humidity and rainfall were recorded (Table 1). Monthly mean temperature values increased gradually from 27.30 and 24.70 °C in April to 30.57 and 31.37 °C in June and July in 2016 and 2017, respectively. In October of the first and second years, the maximum relative humidity was 56.70 and 46.70%, respectively. The first year recorded no rain, and the second year received 5.6 mm.

**Table 1.** Average temperature, relative humidity and rainfall in the study area in Giza, Egypt, during the two growing seasons of 2016 and 2017 *.

| Month | 2016 | | | 2017 | | |
|---|---|---|---|---|---|---|
| | Temperature (°C) | Relative Humidity (%) | Rainfall (mm) | Temperature (°C) | Relative Humidity (%) | Rainfall (mm) |
| April | 27.30 | 38.30 | 0.00 | 24.70 | 41.00 | 1.60 |
| May | 28.90 | 38.67 | 0.00 | 29.27 | 34.00 | 0.00 |
| June | 30.57 | 31.67 | 0.00 | 29.33 | 35.33 | 0.00 |
| July | 30.45 | 46.33 | 0.00 | 31.37 | 42.33 | 2.00 |
| August | 30.47 | 44.33 | 0.00 | 30.85 | 46.33 | 0.00 |
| September | 28.85 | 44.33 | 0.00 | 28.60 | 46.00 | 2.00 |
| October | 27.90 | 56.70 | 0.00 | 26.40 | 46.70 | 0.00 |

* Central Laboratory for Agricultural Climate (CLAC), Agricultural Research Center (ARC), Egypt.

A physical soil analysis was conducted in accordance with Klute [31], and a chemical analysis was carried out in accordance with Page et al. [32]. The soil at the study site is categorized as clay (Table 2). In the first and second years, the soil's pH was 7.21 and 7.41, and its EC was 0.92 and 0.75 dS m$^{-1}$, respectively.

**Table 2.** Soil physical properties and chemical analysis of the experimental site during 2016 and 2017.

| Soil Analysis | 2016 | 2017 |
|---|---|---|
| **Physical Properties** | | |
| Fine Sand (%) | 27 | 21 |
| Silt (%) | 29 | 26 |
| Clay (%) | 44 | 53 |
| Texture | Clay | Clay |
| **Chemical properties** | | |
| pH $_{(1:1)}$ | 7.21 | 7.41 |
| EC $_{(1:1)}$ (dS m$^{-1}$) | 0.92 | 0.75 |
| Organic matter (%) | 2.43 | 2.12 |
| Available N (mg kg$^{-1}$) | 12.3 | 10.7 |
| Available P (mg kg$^{-1}$) | 19.5 | 14.3 |
| Available K (mg kg$^{-1}$) | 76.0 | 91.0 |
| **Irrigation system** | Surface irrigation | Surface irrigation |

### 2.2. Experimental Design and Treatments

The Egyptian cultivar Giza-6 was provided by the Research Institute of Oil Crops, Field Crops Research Institute, Agricultural Research Center (ARC), Egypt. Nigerian cultivars, Samnut-23 and Samnut-24, were obtained from Ahmadu Bello University, Zaria, Nigeria. The American cultivar NC-7 was obtained kindly from the Faculty of Agriculture, Cairo University. The cultivars of peanut used in this investigation were evaluated under three calcium treatments: (1) Gypsum (CaSO$_4$.2H$_2$O) was added when preparing the seedbed at a rate of 1190 kg ha$^{-1}$. (2) Calcium oxide (CaO) was added as a foliar application at a rate of 7.14 kg ha$^{-1}$ twice at 21-day intervals (30 and 51 days after sowing). (3) The control treatment with distilled water as a foliar application was added twice at 21-day intervals.

The experimental design was a randomized complete block in a spilt-plot arrangement with three replications. The main plots were allocated to four peanut cultivars, and sub-plots were devoted to three Ca-forms (soil application in the form of calcium sulfate dihydrate (gypsum), foliar application in the form of calcium oxide and control treatment of distilled water).

### 2.3. Cultural Practices

Peanut seeds were planted in hills by hand in April and May of 2016 and 2017, respectively. In addition, they were thinned to 2 seedling $hill^{-1}$ after full emergence (about two weeks after sowing) to avoid competition between plants. The plot area consisted of four ridges, each 0.60 m apart and 4 m long with an area of 9.6 $m^2$, and were sown on hills with 10 cm spacings. To prevent weeds from growing in the crop field and to break the soil crust, weeding was completed when necessary. Three cultivations were carried out: the first one after establishment, the second one before flowering and the third one during the pegging stage. During the preparation of the seed bed, 360 kg $ha^{-1}$ of super phosphate (15.5% $P_2O_5$), a phosphorus fertilizer, was added. A total of 240 kg $ha^{-1}$ of potassium fertilizer in the form of potassium sulphate (48% $K_2O$) was applied before flowering. Additionally, two equal applications of calcium nitrate (15.5% N) fertilizer at a rate of 240 kg ha-1 each were applied to the soil, the first during sowing and the second 15 days later. Peanut was sprayed twice at the beginning of flowering (40–50 days after sowing) and after a 3-week period with a micronutrients mixture, viz., boron, zinc, iron and manganese at a rate of 0.5 g/L. Cultural practices were carried out in accordance with ARC, the Egyptian Ministry of Agriculture.

### 2.4. Data Collection
### 2.4.1. Peanut Growth and Yield Components Traits

Plant Height (cm) measured from the soil surface up to the top of the main stem, number of branches $plant^{-1}$, number of pegs $plant^{-1}$, number of mature pods $plant^{-1}$, number of seeds $plant^{-1}$, pops $plant^{-1}$, weight of pods $plant^{-1}$ (g), weight of seeds $plant^{-1}$ (g) and plant dry weight (g) were recorded for 20 plants from the inner ridges of each plot at harvest time.

### 2.4.2. Peanut Yields

#### Biological Yield (ton $ha^{-1}$)

The peanut biological yield refers to the total dry matter accumulation of a plant system. The biological yield (ton) per hectare was obtained from the yield of whole plant.

#### Pod Yield (ton $ha^{-1}$)

The total pod yield of the experimental plot was calculated from the three middle ridges of each plot and converted to pod yield, in kilograms.

#### Seed Yield (ton $ha^{-1}$)

The seed yield of the experimental plot was calculated from the yield of pods after separation of the shell and converted to seed yield.

#### Oil Yield (kg $ha^{-1}$)

The oil yield was calculated by multiplying oil percentage by seed yield (kg $ha^{-1}$).

#### Harvest Index (%)

The harvest index represents increased physiological capacity to mobilize photosynthates and translocate them into pod yields. It is calculated by dividing seed yield with biological yield $\times$ 100.



2.4.3. Peanut Quality Traits

Pod Index (Weight of 100 Pods, in Grams)

The weight of 100 sun dried pods (grams) from each plot was taken. The electric balance taken was used.

Seed Index (Weight of 100 Seeds, in Grams)

The weight of 100 sun dried seeds (grams) from each plot was taken. The electric balance taken was used.

Shelling Percentage (%)

Shelling percentage is a measurement of dry matter partitioning between the economically useful part (kernel) and the rest of the fruit (shell), and it is expressed as follows:

$$\text{Shelling percentage} = \frac{\text{weight of seeds}}{\text{weight of pods}} \times 100$$

Pops Percentage (%)

$$\text{Pops percentage} = \frac{\text{number of pops}}{\text{number of pods}} \times 100$$

2.4.4. Peanut Chemical Constitutes

Seed oil percentage, seed calcium percentage and fatty acid composition (FFA) percentage were analyzed. The samples of the analyses were conducted at the Regional Food and Feed Center (RCFF), ARC and at the Chemical Laboratory, Faculty of Agriculture, Cairo University, Egypt.

Seed Oil Percentage (%)

The Soxhlet continuous extraction apparatus was used to determine the oil seed percentage at CLAD according to A.O.A.C, [33].

Free Fatty Acid Percentage (FFA)

In order to determine the FFA percentage, Gas Lipids Chromatography was used, Trace GC Ultra, Thermo Scientific (Waltham, MA, USA) according to Farag et al. [34], at RCFF.

Seed Calcium Percentage

The seed calcium percentage was evaluated via atomic absorption after drying samples at 70 °C until a constant weight was obtained; then, it was ground to a fine powder and sub samples of 0.2 g were taken to be digested using a mixture of sulfuric and perchloric acids [35] at RCFF.

*2.5. Statistical Analysis*

The data of the study were statistically analyzed according to procedures outlined by Steel et al. [36], using the MSTAT-C computer program version 5.5 [37]. Comparing treatment means was conducted using F. test via two way-ANOVA and least significant difference (LSD) at 5% level of probability.

**3. Results and Discussion**

Since there was no significant difference between the two years of the study, the data for the two years were combined.

### 3.1. Peanut Growth and Yield Components Traits

Significant differences among cultivars and calcium applications in the growth and yield component traits were observed (Table 3). The Giza-6 cultivar achieved the highest values of branch number plant$^{-1}$ (17.2), pod number plant$^{1}$ (48.6) and seed number plant$^{-1}$ (61.2). Samnut-24 attained the tallest values of plant height (95.2 cm) and pegs number plant$^{-1}$ (80.9). The foliar calcium application significantly enhanced peanut growth traits: plant height (77.0 cm), branch number per plant (9.8) and pegs number plant$^{-1}$ (82.8), while the soil calcium application significantly improved peanut yield components traits: pod number plant$^{-1}$ (42.5) and seed number plant$^{-1}$ (59.4). The results indicate that the Giza-6 cultivar treated with soil Ca application achieved the highest branch number plant$^{-1}$ (18.0) and pod number plant$^{-1}$ (54.5). The Giza-6, Samnut-24 and NC-7 cultivars treated with foliar Ca application attained the highest seed number plant$^{-1}$ (69.6), the tallest plant height (99.1 cm) and the highest pegs number plant$^{-1}$ (91.0), respectively (Table 3).

**Table 3.** Means of plant height, branch number per plant, peg number per plant, pod number per plant and seed number per plant of peanut cultivars evaluated under calcium treatments and their interactions (combined data of 2016 and 2017 seasons).

| Cultivars | Ca Treatment | Plant Height (cm) | Branch Number Plant$^{-1}$ | Peg Number Plant$^{-1}$ | Pod Number Plant$^{-1}$ | Seed Number Plant$^{-1}$ |
|---|---|---|---|---|---|---|
| Giza-6 | | 67.9 [b] | 17.2 [a] | 56.9 [b] | 48.6 [a] | 61.2 [a] |
| Samnut-23 | | 54.4 [c] | 8.3 [c] | 80.1 [a] | 31.5 [b] | 52.7 [a] |
| Samnut-24 | | 95.2 [a] | 7.0 [c] | 80.9 [a] | 33.6 [b] | 50.8 [a] |
| NC-7 | | 65.0 [b] | 11.0 [b] | 77.1 [a] | 30.0 [b] | 36.8 [b] |
| LSD $_{0.05}$ | | 7.0 | 2.2 | 16.2 | 7.9 | 10.9 |
| | Control | 64.7 [b] | 8.9 [b] | 67.1 [a] | 27.0 [c] | 43.6 [b] |
| | Soil Application (SA) | 75.1 [a] | 9.8 [b] | 74.2 [a] | 42.5 [a] | 59.4 [a] |
| | Foliar Application (FA) | 77.0 [a] | 9.8 [b] | 82.3 [a] | 40.3 [ab] | 54.0 [a] |
| LSD $_{0.05}$ | | 5.6 | 2.0 | 20.6 | 7.5 | 10.1 |
| Giza-6 | Control | 67.3 [d–g] | 13.1 [c–e] | 46.7 [d] | 41.2 [a–e] | 48.5 [b–f] |
| | Soil Application (SA) | 70.6 [c–f] | 18.0 [b] | 56.8 [b–d] | 54.5 [a] | 69.3 [a] |
| | Foliar Application (FA) | 76.6 [cd] | 14.0 [b–d] | 70.8 [b–d] | 53.0 [ab] | 69.6 [a] |
| Samnut-23 | Control | 46.0 [i] | 5.3 [h] | 64.1 [b–d] | 14.5 [g] | 43.8 [b–f] |
| | Soil Application (SA) | 52.1 [hi] | 6.5 [gh] | 114.3 [a] | 43.5 [a–d] | 63.5 [ab] |
| | Foliar Application (FA) | 58.5 [gh] | 8.6 [f–h] | 81.3 [a–d] | 38.8 [b–f] | 52.6 [a–d] |
| Samnut-24 | Control | 80.5 [c] | 6.8 [gh] | 59.7 [bd] | 27.5 [e–g] | 43.0 [c–f] |
| | Soil Application (SA) | 107.0 [a] | 5.8 [h] | 87.3 [a–d] | 38.6 [b–f] | 62.8 [a–c] |
| | Foliar Application (FA) | 99.1 [ab] | 6.6 [gh] | 86.3 [a–d] | 35.6 [c–f] | 51.5 [a–e] |
| NC-7 | Control | 65.3 [e–g] | 9.1 [e–h] | 48.0 [cd] | 24.7 [fg] | 30.6 [f] |
| | Soil Application (SA) | 70.6 [c–f] | 10.1 [d–g] | 88.8 [a–c] | 33.5 [c–f] | 42.1 [d–f] |
| | Foliar Application (FA) | 73.6 [c–e] | 10.1 [d–g] | 91.0 [ab] | 33.8 [c–f] | 42.5 [d–f] |
| LSD $_{0.05}$ | | 11.1 | 4.1 | 41.3 | 15.1 | 20.2 |

Means followed by the same letter within columns are not significantly different at $p = 0.05$, according to the least significant difference test.

Results in Table 4 showed that the Giza-6 cultivar achieved the highest pop number plant$^{-1}$ (11.5), the highest pod weight plant$^{-1}$ (57.9 g), seed weight plant$^{-1}$ (37.9 g) and plant dry weight (200.0 g). The foliar calcium application attained the highest pop number plant$^{-1}$ (6.8) and plant dry weight (148.8 g), while the soil calcium application significantly achieved the highest pod weight plant$^{-1}$ (57.9 g) and seed weight plant$^{-1}$ (37.9 g). The results indicate that the Giza-6 cultivar treated with soil Ca application achieved the highest pod weight plant$^{-1}$ (61.6 g), seed weight plant$^{1}$ (38.3 g) and plant dry weight (233.3 g). The Giza-6 cultivar had the highest pop number plant$^{-1}$ (12.3) under the foliar calcium application.

**Table 4.** Means of pop number per plant, pod weight per plant, seed weight per plant and plant dry weight of peanut cultivars evaluated under calcium treatments and their interactions (combined data of 2016 and 2017 seasons).

| Cultivars | Ca Treatment | Pop Number Plant$^{-1}$ | Pod Weight Plant$^{-1}$ (g) | Seed Weight Plant$^{-1}$ (g) | Plant Dry Weight (g) |
|---|---|---|---|---|---|
| Giza-6 | | 11.5 a | 57.9 a | 37.9 a | 200.0 a |
| Samnut-23 | | 3.0 b | 36.2 b | 25.0 a | 60.0 d |
| Samnut-24 | | 3.1 b | 25.8 c | 18.3 a | 133.3 c |
| NC-7 | | 5.8 b | 38.7 b | 22.9 a | 161.7 b |
| LSD $_{0.05}$ | | 2.9 | 8.8 | 19.8 | 28.1 |
| | Control | 6.0 ab | 31.6 b | 20.0 a | 93.7 c |
| | Soil Application (SA) | 4.1 ab | 44.5 ab | 30.4 a | 146.7 b |
| | Foliar Application (FA) | 6.8 a | 35.8 ab | 22.5 a | 148.8 ab |
| LSD $_{0.05}$ | | 2.3 | 12.9 | 16.4 | 18.3 |
| Giza-6 | Control | 10.8 ab | 43.3 a–c | 30.0 ab | 171.7 bc |
| | Soil Application (SA) | 9.5 a–c | 61.6 ab | 38.3 ab | 223.3 a |
| | Foliar Application (FA) | 12.3 ab | 55.0 a–c | 35.0 ab | 176.7 b |
| Samnut-23 | Control | 4.5 d | 31.6 bc | 20.0 ab | 53.3 d |
| | Soil Application (SA) | 2.0 e | 40.0 a–c | 28.3 ab | 60.0 d |
| | Foliar Application (FA) | 2.6 e | 33.3 b–c | 23.3 ab | 68.3 d |
| Samnut-24 | Control | 2.8 e | 23.3 c | 15.0 ab | 76.6 d |
| | Soil Application (SA) | 2.0 e | 28.3 bc | 21.6 ab | 136.7 c |
| | Foliar Application (FA) | 3.3 e | 23.3 c | 16.6 ab | 171.7 bc |
| NC-7 | Control | 6.1 c–e | 26.6 c | 11.6 b | 73.3 d |
| | Soil Application (SA) | 3.2 e | 48.3 a–c | 33.3 ab | 166.7 bc |
| | Foliar Application (FA) | 9.0 b–d | 33.3 bc | 18.3 ab | 178.3 b |
| LSD $_{0.05}$ | | 4.6 | 34.8 | 34.8 | 36.7 |

Means followed by the same letter within columns are not significantly different at $p$ = 0.05, according to the least significant difference test.

According to the findings of our study, the foliar calcium application significantly improved values of plant height, branch number plant$^{-1}$, plant dry weight, and number of pegs plant$^{-1}$, while the soil calcium application significantly improved peanut yield component traits: pod number per plant, seed number per plant, pod weight per plant and seed weight per plant. This might be attributed to the fact that foliar calcium applications on leaves and stems may affect the physiological and biochemical processes and growth of peanuts. In this context, the physiological and biochemical processes and growth of peanuts may be impacted by the greater calcium concentrations in leaves and stems, which could also indirectly alter the soil's ability to absorb calcium. [23]. Also, the calcium foliar application improved leaf growth and dry weight [24]. However, the foliar application is not seen to be a particularly effective approach in preventing calcium deficiency in peanuts when compared to the use of calcium fertilizer in the soil [15,27].

Using gypsum, the calcium advantages of peanuts are increased. This result may be due to poor availability and absorption of foliar calcium application. Li et al. [23] reported that inorganic foliar calcium fertilizers are frequently used in the production of peanuts. However, calcium absorbed through the foliar has low availability. Through the phloem pathway, calcium cannot be redistributed from older to younger plant tissues. [10]. In agreement with our findings, Rahman [38] demonstrated that the application of Ca had significant effect on the peanut growth. The applying of Ca significantly increased in germination rate and promoted the growth of groundnut [14,15]. Ca application significantly improved shoot length and dry weight [4]. Ullah et al. [39] found that the application of gypsum increased the growth of peanut under rainfed conditions. Safarzadeh Vishkaee [40] indicated that the application of gypsum as a source of Ca has produced the large pods.

### 3.2. Peanut Yields

Significant differences among the treatments in the biological, pods, seed, oil yields and harvest index were observed (Table 5). The Samnut-24 cultivar achieved the highest values of biological, seed and oil yields. The NC-7 cultivar attained the highest value of pod yield. The calcium application significantly enhanced peanut biological, pod, seed, oil yields and harvest index in four cultivars compared to no additional Ca. The results indicate that the NC-7 cultivar treated with soil Ca application resulted in the maximum values of biological yield (92.9-ton ha$^{-1}$), pod yield (6.8-ton ha$^{-1}$), seed yield (4.4-ton ha$^{-1}$) and oil yield (2247.0 kg ha$^{-1}$), while the Samnut-23 cultivar treated with soil calcium applying resulted in the highest harvest index (14.7%).

**Table 5.** Means of biological yield per hectare, pod yield per hectare, seed yield per hectare, oil yield per hectare and harvest index of peanut cultivars evaluated under calcium treatments and their interactions (combined data of 2016 and 2017 seasons).

| Cultivar | Ca Treatment | Biological Yield (ton ha$^{-1}$) | Pod Yield (ton ha$^{-1}$) | Seed Yield (ton ha$^{-1}$) | Oil Yield (kg ha$^{-1}$) | Harvest Index (%) |
|---|---|---|---|---|---|---|
| Giza-6 | | 65.6 [b] | 4.6 [b] | 2.9 [b] | 1481 [b] | 7.5 [b] |
| Samnut-23 | | 26.9 [c] | 3.2 [c] | 1.9 [c] | 952.7 [c] | 12.2 [a] |
| Samnut-24 | | 80.4 [a] | 5.0 [ab] | 3.4 [a] | 1739 [a] | 6.4 [b] |
| NC-7 | | 71.5 [ab] | 5.2 [a] | 3.3 [a] | 1728 [a] | 8.8 [b] |
| LSD $_{0.05}$ | | 9.7 | 0.4 | 2.6 | 149.8 | 2.5 |
| | Control | 48.1 [b] | 3.5 [b] | 2.2 [c] | 1127 [c] | 7.5 [a] |
| | Soil Application (SA) | 70.2 [a] | 5.5 [a] | 3.5 [a] | 1779 [a] | 9.4 [a] |
| | Foliar Application (FA) | 64.0 [a] | 4.8 [a] | 2.7 [b] | 1636 [a] | 9.1 [a] |
| LSD $_{0.05}$ | | 8.4 | 0.4 | 3.7 | 191.6 | 2.4 |
| Giza-6 | Control | 51.4 [e] | 3.6 [fg] | 2.4 [de] | 1231 [de] | 7.6 [cd] |
| | Soil Application(SA) | 66.3 [b–e] | 5.7 [b] | 3.5 [bc] | 1818 [bc] | 9.2 [b–d] |
| | Foliar Application (FA) | 79.4 [a–c] | 5.2 [bc] | 2.3 [de] | 1757 [bc] | 7.0 [d] |
| Samnut-23 | Control | 21.1 [f] | 2.0 [h] | 1.2 [f] | 603.8 [f] | 10.1 [a–d] |
| | Soil Application (SA) | 28.9 [f] | 4.0 [e–g] | 2.3 [de] | 1180 [de] | 14.7 [a] |
| | Foliar Application (FA) | 25.2 [f] | 3.2 [g] | 1.9 [ef] | 1077 [e] | 13.3 [ab] |
| Samnut-24 | Control | 59.8 [de] | 3.6 [fg] | 2.3 [de] | 1194 [de] | 6.2 [d] |
| | Soil Application (SA) | 92.7 [a] | 5.4 [bc] | 3.7 [ab] | 1869 [ab] | 5.9 [d] |
| | Foliar Application (FA) | 87.2 [a] | 5.6 [b] | 3.9 [ab] | 1987 [ab] | 6.7 [d] |
| NC-7 | Control | 60.0 [de] | 4.3 [c–e] | 2.9 [cd] | 1480 [cd] | 11.9 [a–c] |
| | Soil Application (SA) | 92.9 [a] | 6.8 [a] | 4.4 [a] | 2247 [a] | 7.8 [cd] |
| | Foliar Application (FA) | 64.1 [c–e] | 5.0 [b–d] | 2.9 [cd] | 1721 [bc] | 9.3 [b–d] |
| LSD $_{0.05}$ | | 16.9 | 0.8 | 7.4 | 383.1 | 4.8 |

Means followed by the same letter within columns are not significantly different at *p* = 0.05, according to the least significant difference test.

At the calcium soil application, pod yield significantly increased by 55.82, 94.23, 50.27 and 57.79%, respectively, for the Giza-6, Samnut-23, Samnut-24 and NC-7 cultivars compared to no additional calcium. Seed yield significantly increased by 46.72, 90.16, 56.77 and 51.54%, respectively, for the Giza-6, Samnut-23, Samnut-24 and NC-7 cultivars when the peanut plants were treated with calcium soil application compared to no additional calcium. Furthermore, the oil yield significantly increased by 47.68, 95.42, 56.36 and 51.82%, respectively, for the Giza-6, Samnut-23, Samnut-24 and NC-7 cultivars under calcium soil application compared to no additional calcium.

Under the calcium foliar application, pod yield significantly increased by 42, 57.21, 55.76 and 15.59% for the Giza-6, Samnut-23, Samnut-24 and NC-7 cultivars, respectively, compared to no additional calcium. Seed yield increased by 61.47 and 66.52%, respectively, for the Samnut-23 and Samnut-24 cultivars and decreased by −3.68 and −0.34%, respectively, for the Giza-6 and NC-7 when the peanut plants were treated with a calcium foliar application compared to no additional calcium. Also, the oil yield increased by 42.72, 78.37,

66.41 and 16.14%, respectively, for the Giza-6, Samnut-23, Samnut-24 and NC-7 cultivars under a calcium foliar application compared to the control (Table 5).

A soil calcium application significantly improved peanut production compared to a foliar calcium application. The findings of the current experiment showed that all the four peanut cultivars responded differently to the application of calcium fertilizers. Our results indicated that the application of calcium fertilizers is an important factor for increasing peanut yield. This result might be due to the soil calcium application using gypsum to make calcium possible in surrounding soil, and peanut pod calcium needs are taken from the soil. Similar findings have been reported by Pegues et al. [18], who found that gypsum is supplemental Ca in the fruiting area during mid-season growth but was diluted by the time of harvest. The gypsum application attained a higher peanut pod yield compared to the zero-gypsum application due to the adequate availability of Ca in the fruiting zone [41]. Ca is distributed within the plant, and the flow of uptake is from the root surface to xylem, then to the shoot [5]. Further, via direct diffusion, Ca enters into the seed from the soil via the hull [42]. Most of the pod calcium needs are taken directly from soil, through the shell of pods, rather than via the roots, and then down through the peg [22].

The NC-7 cultivar with the treatment soil application achieved the maximum values of pod, seed and oil yields $ha^{-1}$. The result might be due to the application of calcium that improved the pod yield. It is thought that increasing number of the mature pods and pod weight per plant with this application of calcium were the main factors that were effective on pod yield. Also, the application of calcium attained the highest value of seed yield compare to the control [28], in addition, the oil content enhanced with the increase in the calcium application. It was likely due to the increase in photosynthetic materials because the seed oil fully depends on the production of these materials after flowering [40,43,44].

Our findings are in line with the beneficial role of gypsum as a soil amendment on peanut yield. These findings can be confirmed by Singh et al. [45], who found that the dose of gypsum at 250–500 kg $ha^{-1}$ achieved the highest value of pod yield. The gypsum application at a rate of 200–500 kg $ha^{-1}$ significantly improved peanut yield [46]. Gypsum significantly enhanced the pod yield [47]. Also, gypsum at a rate of 500 kg per acre significantly increased the pod yield [48]. The addition of gypsum at rates of 0.5 and 1.0 tons per acre increased biological yield, pod yield, and seed yields [49]. The application of gypsum significantly increased seed oil content [50]. Increasing gypsum application from 0 to 400 kg $ha^{-1}$ significantly increased oil yield $ha^{-1}$ [51].

### 3.3. Peanut Quality Traits

Significant differences among cultivars and calcium applications in peanut quality traits were observed (Table 6). The NC-7 cultivar attained the highest values of pod index (180.9 g) and seed index (78.7 g). The Samnut-24 attained the highest shelling percentage (68.1%), while the Giza-6 cultivar had a higher pops percentage (23.1%). The soil calcium application significantly enhanced pod index (140.0 g) and seed index (61.42 g), while the foliar calcium application had a higher shelling percentage (66.6%). On the other hand, the control (no-additional Ca) attained the highest pops percentage (23.1). The results indicated that the NC-7 cultivar achieved the highest pod index (203.2 g) and the highest seed index (84.1 g) under soil Ca application and the highest pops percentage (38.6) under the control (no additional Ca), while the Samnut-24 cultivar treated with a soil Ca application resulted in the highest shelling percentage (67.7%).

**Table 6.** Means of pod index, seed index, shelling percentage and pops percentage of peanut cultivars evaluated under calcium treatments and their interaction (combined data of 2016 and 2017 seasons).

| Cultivars | Ca Treatment | Pod Index (g) | Seed Index (g) | Shelling Percentage | Pops Percentage |
|---|---|---|---|---|---|
| **Giza-6** | | 157.9 [b] | 70.6 [b] | 64.6 [b] | 23.1 [a] |
| **Samnut-23** | | 109.2 [c] | 48.9 [c] | 60.3 [c] | 9.7 [b] |
| **Samnut-24** | | 85.1 [d] | 38.7 [d] | 68.1 [a] | 11.4 [b] |
| **NC-7** | | 180.9 [a] | 78.7 [a] | 64.2 [b] | 21.7 [a] |
| **LSD [0.05]** | | 12.0 | 2.5 | 3.2 | 7.5 |
| | **Control** | 128.1 [b] | 58.0 [b] | 62.5 [b] | 20.5 [a] |
| | **Soil Application (SA)** | 140.0 [a] | 61.4 [a] | 63.1 [ab] | 14.1 [b] |
| | **Foliar Application (FA)** | 134.0 [ab] | 58.1 [b] | 66.6 [a] | 14.5 [b] |
| **LSD [0.05]** | | 8.8 | 2.7 | 3.6 | 5.5 |
| **Giza-6** | **Control** | 148.3 [d] | 68.0 [e] | 62.2 [b–d] | 26.6 [b] |
| | **Soil Application (SA)** | 167.2 [bc] | 73.1 [c–e] | 67.1 [ab] | 18.8 [b–e] |
| | **Foliar Application (FA)** | 165.0 [b0–d] | 69.6 [de] | 66.3 [a–c] | 22.7 [bc] |
| **Samnut-23** | **Control** | 103.3 [ef] | 47.1 [f] | 56.7 [d] | 10.2 [d–f] |
| | **Soil Application (SA)** | 104.5 [ef] | 48.8 [f] | 59.5 [cd] | 8.0 [ef] |
| | **Foliar Application (FA)** | 116.2 [e] | 48.5 [f] | 67.2 [ab] | 7.9 [ef] |
| **Samnut-24** | **Control** | 81.0 [g] | 36.8 [g] | 65.2 [a–c] | 8.8 [d–f] |
| | **Soil Application (SA)** | 85.5 [g] | 39.5 [g] | 67.7 [ab] | 8.7 [d–f] |
| | **Foliar Application (FA)** | 83.3 [g] | 38.0 [g] | 66.9 [ab] | 13.5 [c–f] |
| **NC-7** | **Control** | 171.5 [b] | 74.0 [cd] | 60.6 [b–d] | 38.6 [a] |
| | **Soil Application (SA)** | 203.2 [a] | 84.1 [a] | 63.5 [b–d] | 18.5 [b–f] |
| | **Foliar Application (FA)** | 177.0 [b] | 77.1 [bc] | 66.1 [a–c] | 13.8 [c–f] |
| **LSD [0.05]** | | 17.6 | 5.4 | 7.3 | 11.0 |

Means followed by the same letter within columns are not significantly different at $p = 0.05$, according to the least significant difference test.

It is clear that the utilization of calcium had a prominent impact on pod index, seed index, pops percentage and shelling percentage. These results might be due to the fact that calcium is an important factor that was effective in peanut pops, and the deficiency of Ca in the fruiting zone resulted in the pegs forming very few pods. The application of calcium on peanuts can be beneficial and increase seed quality. Thus, applying calcium at 45 days after sowing was the most effective for pod development [52]. Also, Kamara et al. [28] reported that the highest filled pods were obtained with the calcium treatment. In addition, the dose of gypsum at 400 kg ha$^{-1}$ significantly increased the quality traits of peanut [47]. The shelling percentage has been significantly increased by gypsum application [48]. Furthermore, Ghosh et al. [53] indicated that seed treated with gypsum achieved the lowest number of immature pod plant $^{-1}$. Yang et al. [15] reported that soil calcium deficiency produces empty pods.

*3.4. Peanut Chemical Constituents*

Significant differences among cultivars and Ca levels in the chemical composition of seeds were observed (Table 7). The application of Ca increased seed oil, free fatty acids and seed calcium percentages for all cultivars. The results indicated that the NC-7 cultivar achieved a higher seed oil percentage (50.0%) and free fatty acids percentage (0.741%). The Samnut-24 cultivar had a higher seed calcium percentage (8.9%). The soil calcium application significantly improved seed oil percentage (50.6%) and seed calcium percentage (8.3%), while the foliar calcium application had a higher free fatty acids percentage (0.7583%). The Giza-6 cultivar sprayed with Ca application had the highest free fatty acids percentage (0.8167%), while the Samnut-35 cultivar treated with soil Ca application achieved the highest seed oil percentage (51.3) and seed calcium percentage (9.5).

**Table 7.** Means of seed oil, free fatty acid and seed calcium percentages of peanut cultivars evaluated under calcium treatments and their interactions (combined data of 2016 and 2017 seasons).

| Cultivar | Ca Treatment | Seed Oil Percentage | Free Fatty Acid Percentage | Seed Calcium Percentage |
|---|---|---|---|---|
| **Giza-6** | | 49.5 [a] | 0.53 [c] | 6.3 [d] |
| **Samnut-23** | | 49.5 [a] | 0.64 [b] | 8.4 [b] |
| **Samnut-24** | | 49.8 [a] | 0.52 [c] | 8.9 [a] |
| **NC-7** | | 50.0 [a] | 0.74 [a] | 7.3 [c] |
| **LSD [0.05]** | | 1.1 | 0.02 | 0.9 |
| | **Control** | 48.5 [b] | 0.37 [c] | 6.6 [d] |
| | **Soil Application (SA)** | 50.6 [a] | 0.57 [b] | 8.3 [a] |
| | **Foliar Application (FA)** | 49.9 [a] | 0.75 [a] | 7.9 [c] |
| **LSD [0.05]** | | 1.0 | 0.02 | 0.1 |
| **Giza-6** | **Control** | 47.5 [d] | 0.31 [l] | 6.0 [l] |
| | **Soil Application (SA)** | 50.7 [a] | 0.65 [f] | 6.6 [i] |
| | **Foliar Application (FA)** | 50.2 [a–c] | 0.81 [c] | 6.3 [j] |
| **Samnut-23** | **Control** | 48.2 [cd] | 0.33 [kl] | 7.4 [g] |
| | **Soil Application (SA)** | 50.4 [ab] | 0.40 [ij] | 8.5 [de] |
| | **Foliar Application (FA)** | 50.0 [a–c] | 0.73 [de] | 8.8 [c] |
| **Samnut-24** | **Control** | 48.4 [b–d] | 0.38 [i–k] | 7.5 [g] |
| | **Soil Application (SA)** | 51.3 [a] | 0.50 [g] | 9.5 [a] |
| | **Foliar Application (FA)** | 49.5 [a–d] | 0.78 [cd] | 8.6 [b] |
| **NC-7** | **Control** | 49.8 [a–c] | 0.48 [gh] | 5.6 [m] |
| | **Soil Application (SA)** | 50.1 [a–c] | 0.75 [de] | 8.4 [e] |
| | **Foliar Application (FA)** | 50.0 [a–c] | 0.70 [ef] | 8.0 [f] |
| **LSD [0.05]** | | 2.1 | 0.05 | 0.1 |

Means followed by the same letter within columns are not significantly different at $p = 0.05$, according to the least significant difference test.

The addition of calcium had a significant effect on seed oil, FFA and seed calcium percentages. The NC-7 and Samnut-24 cultivars attained the maximum values of seed oil, FFA and seed calcium percentages. The result might be due to the gypsum application that affected the size of the seed and pod, as well as increased dry matter that included oil bodies. The positive properties of gypsum and its high solubility in the soil and the presence of elements such as calcium and sulfur in its structure could cause an increase in the oil content with increasing calcium levels that were provided by gypsum [40,53,54]. Furthermore, Arnold et al. [55] reported that the gypsum fertilization achieved the highest seed calcium content.

## 4. Conclusions

We can conclude that the application of calcium fertilizers is a substantial factor for enhancing peanut production and improving seed quality. Calcium application significantly enhanced peanut growth, yield components, biological, pod, seed, oil yields, seed oil, free fatty acids and seed calcium percentages in four cultivars, compared to the control. The soil calcium application significantly improved peanut production compared to foliar calcium application. The NC-7 cultivar treated with a soil Ca application achieved the highest values of biological yield (92.9-ton ha$^{-1}$), pod yield (6.8-ton ha$^{-1}$), seed yield (4.4-ton ha$^{-1}$), oil yield (2247.0 kg ha$^{-1}$), pod index (203.2 g) and seed index (84.1 g) under clay soil conditions. The interaction between the NC-7 cultivar and soil calcium applications is recommended as it attained the best combination, leading to the highest yield and seed quality of peanut.

**Author Contributions:** Conceptualization, M.A.; data acquisition, M.H. and S.S.; design of methodology, M.H. and S.S.; writing and editing, M.A. and F.H.G. All authors have read and agreed to the published version of the manuscript.

**Funding:** This work was funded by the Deanship of Scientific Research at Jouf University through the Fast-track Research Funding Program.

**Data Availability Statement:** All data is included in the manuscript.

**Conflicts of Interest:** The authors declare no conflict of interest.

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
