# Peer review of "Response of Diverse Peanut Cultivars to Calcium Fertilization under Clay Soil Conditions"

_agronomy, doi:10.3390/agronomy13102656_

Round 1
Reviewer 1 Report
The manuscript provides some interesting scientific information on the effects of calcium fertilization on the growth and yield of peanut under clay soil condition. The referee thinks that this manuscript has a merit to be issued in Agronomy; however, the manuscript also has several deficiencies to be improved before acceptance for publication.
As the title, also line 99, suggested, one of the measure issues in this study would be the effectiveness of calcium application to the growth of peanut ‘under clay soil condition’. The authors compared the results of this study with those of prior researches conducted under different types of soil, however, discussion on this topic was lacking.
Line 267: Ullah et al. also agree with our findings that the growth of groundnut increase but cease when applying gypsum.
How could you explain this conflict between your results and those of Ullah’s study?
Line 310: the gypsum is possible that there was short-term availability of supplemental Ca in the pegging zone during mid-season growth, but was diluted by the time of harvest.
You applied gypsum to the soil during the field preparation (line 130). Could you say the gypsum you applied was fairly effective to the pod development, since peg penetration to the soil might be taken place around 40~50 days after seed sowing? Why did you apply gypsum during the flowering stage?
Line 120 respectively. Period was missing.
Table 2 Include the information on the amount of exchangeable calcium in the soil, if you have.
Line 131 In which growth stage did you applied calcium to the plants? You only mentioned ‘twice at 21-day intervals’.
Line 148 ‘/hectare’ to ‘ha-1’
Line 154 The abbreviation of Agricultural Research Center ‘ARC’ had already been explained in line 125.
Line 180and line 183 Are the terms ‘Pod Index’ and ‘Seed Index’ commonly used for peanut? ‘100-pod weight’ and ‘100-grain (or seed) weight’ would be common.
Line 200 ‘(A.O.A.C,[33].’ to ‘A.O.A.C. [33].’
Line 230 ‘Table (4)’ to ‘Table 4’
Table 3 to 7 Effective digit should be one decimal place (except ‘Free fatty acid percentage’ in table 7; two decimal place should be adequate).
Line 276 to 284 The font is bold.
Line 280 The control plant was described as ‘zero Ca’. But, you applied Ca by using calcium nitrate for base fertilizer. It would be more appropriate to describe ‘zero Ca’ as ‘no-additional Ca’.
Line 307: fruiting zone The words ‘pegging zone’ was used in line 69. Are the ‘fruiting zone’ and ‘pegging zone’ different area in the soil?
Line 325: The result might be due to application of calcium at the time of flowering increased the pod yield.
You applied gypsum to the soil during seed bed preparation (line 130).
Line 340: The application of gypsum at the rate of 0.5 and 1.0-ton gypsum per fed increased …
Check the text.
Line 347 ‘Table 6’?
Line 350 140.0 g
Line 362 Duplication of ‘pops percentage’.
Author Response
Reviewer (1)
|
Comment |
Action |
|
As the title, also line 99, suggested, one of the measure issues in this study would be the effectiveness of calcium application to the growth of peanut ‘under clay soil condition’. The authors compared the results of this study with those of prior researches conducted under different types of soil, however, discussion on this topic was lacking. |
Done, the discussion has been improved |
|
Line 267: Ullah et al. also agree with our findings that the growth of groundnut increase but cease when applying gypsum. How could you explain this conflict between your results and those of Ullah’s study? |
The objective of Ullah’s study was to study the effect of gypsum application on groundnut growth and nodules under rain fed condition but our study interested to investigate the influence of calcium (Ca) applications on production and seeds quality of four diverse peanut cultivars from different regions; USA (NC-7), Egypt (Giza-6) and Nigeria (Samnut-23 and Samnut-24) under clay soil conditions with surface irrigation |
|
Line 310: the gypsum is possible that there was short-term availability of supplemental Ca in the pegging zone during mid-season growth, but was diluted by the time of harvest. You applied gypsum to the soil during the field preparation (line 130). Could you say the gypsum you applied was fairly effective to the pod development, since peg penetration to the soil might be taken place around 40~50 days after seed sowing? Why did you apply gypsum during the flowering stage? |
To enhance the availability and activity of Ca as an essential nutrient which playing an important role in peanut pod formation. The growing season for peanuts can be divided into three distinctive phases; pre-bloom/bloom, pegging/pod set and pod fill/maturity. Because the flowering started after 40 days after sowing for tested cultivars. After that the peanut need Ca to facilitate fertilization and to help pods to mature without any empty pods (pops).
|
|
Table 2 Include the information on the amount of exchangeable calcium in the soil, if you have. |
Thank you, soil physical properties and chemical analysis have been done as showing in table 2 but the exchangeable calcium in the soil was not analyzed |
|
Line 120 respectively. Period was missing. |
Done, the period has been added and marked with red color. |
|
Line 131 In which growth stage did you applied calcium to the plants? You only mentioned ‘twice at 21-day intervals’. |
Done, the period has been added and marked with red color. |
|
Line 148 ‘/hectare’ to ‘ha-1’ |
Done, units have been changed and marked with red color. |
|
Line 154 The abbreviation of Agricultural Research Center ‘ARC’ had already been explained in line 125. |
Done, the abbreviation has been changed and marked with red color. |
|
Line 180and line 183 Are the terms ‘Pod Index’ and ‘Seed Index’ commonly used for peanut? ‘100-pod weight’ and ‘100-grain (or seed) weight’ would be common. |
Done, Yes, the pod and seed index are commonly used for peanut and there is no different between pod index and 100-pod weight. Also, seed index and 100-seed weight as following: Pod Index (weight of 100-pod, in grams): weight of 100 sun dried pods (grams) from each plot was taken. Seed Index (weight of 100-seed, in grams): weight of 100 sun dried seeds (grams) from each plot was taken. Marked with red color |
|
Line 200 ‘(A.O.A.C,[33].’ to ‘A.O.A.C. [33].’ |
Done, the reference has been changed and marked with red color. |
|
Line 230 ‘Table (4)’ to ‘Table 4’ |
Done, the table number has been changed and marked with red color. |
|
Table 3 to 7 Effective digit should be one decimal place (except ‘Free fatty acid percentage’ in table 7; two decimal place should be adequate). |
Done, the tables from 3 to 7 have been updated one decimal expect Free fatty acid percentage, two decimal and marked with red color. |
|
Line 276 to 284 The font is bold. |
Done, the font has been changed and marked with red color. |
|
Line 280 The control plant was described as ‘zero Ca’. But, you applied Ca by using calcium nitrate for base fertilizer. It would be more appropriate to describe ‘zero Ca’ as ‘no-additional Ca’. |
Done, the zero Ca has been changed to no-additional Ca in whole manuscript and marked with red color. |
|
Line 307: fruiting zone the words ‘pegging zone’ was used in line 69. Are the ‘fruiting zone’ and ‘pegging zone’ different area in the soil? |
Done, there is no different between fruiting zone’ and ‘pegging zone. However, the growing season for peanuts can be divided into three distinctive phases; pre-bloom/bloom, pegging/pod set and pod fill/maturity. |
|
Line 325: The result might be due to application of calcium at the time of flowering increased the pod yield. You applied gypsum to the soil during seed bed preparation (line 130). |
The soil calcium application using gypsum provided the adequate availability of Ca in the fruiting zone and peanut pods calcium needs are taken from the soil. Gypsum delivers Ca to the fruiting zone, which makes Ca available during kernel and pod development. Also. gypsum is possible that there was short-term availability of supplemental Ca in the pegging zone during mid-season growth, but was diluted by the time of harvest. Marked with red color lines: 248-255 |
|
Line 340: The application of gypsum at the rate of 0.5 and 1.0-ton gypsum per fed increased … Check the text. |
Done, the text has been checked, corrected and marked with red color. |
|
Line 347 ‘Table 6’? |
Done, the table number has been corrected and marked with red color. |
|
Line 350 140.0 g |
Done, the number has been corrected and marked with red color. |
|
Line 362 Duplication of ‘pops percentage’ |
Done, the duplication has been corrected and marked with red color. |

Reviewer 2 Report
The language needs a significant improvement and many phrasing errors are stated through the manuscript.
For example,
line 26, ‘vale’ à’value’;
line 28, ‘cultivar soil’à’ cultivar and soil’;
line 33, ‘Is’à ‘It is’;
line 35-37, the sentence ‘The peanut is a, ……monly cultivated’, need to be rephrased;
line 57, ‘suffer’à’suffering’;
line 92, ‘face’ à’facing’;
line 248-249, never repeat the sentence. ’However, compared with …… avoid calcium deficiency in peanuts’. Try to state it in a different way.
Why do you choose these 4 cultivars to conduct the experiment, are they mostly cultivated and produced in your areas? what is the practically guiding significances of the study for farmers to improve their production? These need to be clarified in the introduction section.
The language of the manuscript should be carefully revised by an English native speaker.
Author Response
Reviewer (2)
|
Comment |
Action |
|
The language of the manuscript should be carefully revised by an English native speaker |
Done, the English language has been improved |
|
line 26, ‘vale’ à’value’; |
Done, value has been corrected and marked with red color. |
|
line 28, ‘cultivar soil’à’ cultivar and soil’; |
Done, cultivar soil has been corrected and marked with red color. |
|
line 33, ‘Is’à ‘It is’; |
Done, Is have been corrected and marked with red color. |
|
line 35-37, the sentence ‘The peanut is a, ……monly cultivated’, need to be rephrased; |
Done, the sentence has been rephrased and marked with red color. |
|
line 57, ‘suffer’à’suffering’; |
Done, suffer has been corrected and marked with red color. |
|
line 92, ‘face’ à’facing’ |
Done, face has been corrected and marked with red color. |
|
line 248-249, never repeat the sentence. ’However, compared with …… avoid calcium deficiency in peanuts’. Try to state it in a different way. |
Done, the sentence has been rephrased and marked with red color. |
|
Why do you choose these 4 cultivars to conduct the experiment, are they mostly cultivated and produced in your areas? |
Done, In Egypt, peanut is considered the first oil crop concerning the cultivated area. Also, the gap of oil production is 90 %. These four diverse peanut cultivars namely; NC-7, Giza-6, Samnut-23 and Samnut-24 may be used to make the country self-sufficient in edible oil, it is extremely necessary to increase the total production of peanut. |
|
what is the practically guiding significances of the study for farmers to improve their production?
|
In this study, we will address the role of Ca applications in peanut under clay soil conditions. Little is known about the performance of cultivars Giza-6, Samnut-23, Samnut-24 and NC-7 grown in clay soil under calcium deficiency. For farmers, NC-7 cultivar treated with soil Ca application achieved the highest values of biological yield, pod yield, seed yield, oil yield, pod index and seed index under clay soil conditions |
|
These need to be clarified in the introduction section. |
Done, the sentence has been improved and marked with red color. |
Reviewer 3 Report
Peanut is one of the most important cash legume crop grown mainly for its edible seeds, it has important research value. The habitat characteristics such as the temperature, rainfall, soil texture are different in various regions of the world, so it is necessary to conduct targeted research on peanut. In this study, the effects of soil application in the form of calcium sulfate dihydrate, foliar application in the form of calcium oxide on four peanut varieties were conducted, and a series of data were measured. Results indicated that NC-7 cultivar with soil calcium applications was the optimal combination. The manuscript is straightforward and not innovative.
Other suggestions:
1.The introduction is too much about basic knowledge, such as the second paragraph, the function of calcium, and more need to grasp the progress of research and scientific questions.
2.Statistical problem. Check the results carfully, such as Seeds number in Table 3, for Cultivars factor, lsd0.05 was 10.93, howerver, the seeds number of Giza-6 (61.21) was not significant more than NC-7(36.83). Also in table 3, for Ca-treatment factor, the seeds number of the three treatments were c, a, ab, they should be b, a, a?
3.Results and discussions should be written separately.
4. Format problem. Check the format carefully, such as Lines 276-284 are not bold, the format of the references is not uniform.
Author Response
Reviewer (3)
|
Comment |
Action |
|
The introduction is too much about basic knowledge, such as the second paragraph, the function of calcium, and more need to grasp the progress of research and scientific questions. |
Done, the introduction has been improved and marked with red color. |
|
Statistical problem. Check the results carfully, such as Seeds number in Table 3, for Cultivars factor, lsd0.05 was 10.93, howerver, the seeds number of Giza-6 (61.21) was not significant more than NC-7(36.83). Also in table 3, for Ca-treatment factor, the seeds number of the three treatments were c, a, ab, they should be b, a, a? |
Done, the statistical letters have been improved and marked with red color. |
|
Results and discussions should be written separately. |
The journal format allowed to write the results combined with discussion. as following: https://www.mdpi.com/journal/agronomy/instructions |
|
Format problem. Check the format carefully, such as Lines 276-284 are not bold, |
Done, the manuscript has been checked and corrected according to journal format |
|
the format of the references is not uniform. |
Done, the references have been checked and corrected according to journal format |
Reviewer 4 Report
Dear Authors,
General comments
1. The article has sufficient data.
2. The purpose of the study is not well defined. Is the soil deficient in Ca? Where is the initial available Ca value? pH is more than 7.0. There should be post-harvest soil test data (whether it increased the alkalinity of the soil further?).
3. Please check the reviewed manuscript.
4. The English language is poor with so many grammatical errors.
Title: Please change the title. It may be:
"Impact of Calcium Fertilization on Peanut Cultivars under Clay Soil Conditions"
Follow the MDPI style: CAPITALIZE THE FIRST LETTER OF EACH IMPORTANT WORDS (except prepositions)
Abstract: Abstract is fine; however, the interaction effect between cultivar and Ca should be mentioned.
Introduction: Comments made in the reviewed manuscript (PDF file attached). Please do the needful.
M&M: This part has some limitations. Address all the issues raised properly.
Results and Discussion: This is the weakest part. You should separate results and discussion. Write results based on the statistical analysis you have done. Highlighting the treatment with the highest value is not the right procedure for narration of the result.
Conclusion: Recast. If you describe the results and discussion part properly, the conclusion will be easier. Also, mention the recommendations of the study clearly.
Recommendation: It can be further considered after major revision.
References: Please follow the journal style.
All the best!!!

The English language is poor with so many grammatical errors.
Author Response
Reviewer (4)
|
Comment |
Action |
|
The article has sufficient data. |
Thank you |
|
The purpose of the study is not well defined. Is the soil deficient in Ca? Where is the initial available Ca value? pH is more than 7.0. There should be post-harvest soil test data (whether it increased the alkalinity of the soil further?). |
Done, the word has been changed |
|
Please check the reviewed manuscript. |
Done, the reviewed manuscript has been checked |
|
The English language is poor with so many grammatical errors. |
Done, the English language has been improved |
|
Title: Please change the title. It may be: "Impact of Calcium Fertilization on Peanut Cultivars under Clay Soil Conditions" Follow the MDPI style: CAPITALIZE THE FIRST LETTER OF EACH IMPORTANT WORDS (except prepositions) |
Thank you for your suggested title. Our title revealed the objective the study that; to investigate the effect of soil and foliar applications of calcium on the productivity and seed quality of four diverse peanut cultivars from different regions; USA (NC-7), Egypt (Giza-6) and Nigeria (Samnut-23 and Samnut-24) under clay soil conditions Done, the title has been improved according to MDPI style and marked with red color
|
|
Abstract: Abstract is fine; however, the interaction effect between cultivar and Ca should be mentioned. |
Done, the abstract has been improved and the interaction between cultivar and Ca marked with red color |
|
Introduction: Comments made in the reviewed manuscript (PDF file attached). Please do the needful. |
Done, the reviewed manuscript has been checked and needed changes have been done |
|
M&M: This part has some limitations. Address all the issues raised properly. |
Done, the materials and methods has been improved and marked with red color |
|
Results and Discussion: This is the weakest part. You should separate results and discussion. Write results based on the statistical analysis you have done. Highlighting the treatment with the highest value is not the right procedure for narration of the result. |
The journal format allowed to write the results combined with discussion. as following: https://www.mdpi.com/journal/agronomy/instructions |
|
Conclusion: Recast. If you describe the results and discussion part properly, the conclusion will be easier. Also, mention the recommendations of the study clearly. |
Done, the conclusion has been improved and marked with red color |
|
References: Please follow the journal style. |
Done, the references have been checked and corrected according to journal format |
Round 2
Reviewer 2 Report
No more suggestions.
Author Response
|
Comment |
Action |
|
No more suggestions. |
Thank you |
Reviewer 3 Report
The manuscript has been improved greatly after revision.
Author Response
|
Comment |
Action |
|
The manuscript has been improved greatly after revision. |
Thank You |
Reviewer 4 Report
Dear authors,
I must appreciate that you have done enough exercise in improving the manuscript.
Please look into the English language once again.
Also, change the keywords. I have attached the corrected pdf version.
Thanks.

Minor correction is needed.
Author Response
|
Comment |
Action |
|
must appreciate that you have done enough exercise in improving the manuscript. |
Thank you |
|
Please look into the English language once again. |
Done, the English language has been improved |
|
Also, change the keywords. I have attached the corrected pdf version. |
Done, the keywords have been changed and marked with purple color |
|
Please write: Pod yield, Oil yield, quality |
Done, Pod yield, Oil yield, quality have been added and marked with purple color |
|
Delete year |
Done, the word of year has been deleted |
|
Put * below the table also |
Done, * has been added below the table and marked with purple color |
|
year |
Done, year has been improved |
|
Delete seasons |
Done, seasons have been deleted |
|
Delete season |
Done, season has been deleted |